# Matrix Optimization of Ultra High Performance Concrete for Improving Strength and Durability

**DOI:** 10.3390/ma14226944

**Published:** 2021-11-17

**Authors:** Julio A. Paredes, Jaime C. Gálvez, Alejandro Enfedaque, Marcos G. Alberti

**Affiliations:** Departamento de Ingeniería Civil: Construcción, E.T.S de Ingenieros de Caminos, Canales y Puertos, Universidad Politécnica de Madrid, c/Profesor Aranguren, s/n, 28040 Madrid, Spain; julparedes@gmail.com (J.A.P.); alejandro.enfedaque@upm.es (A.E.); marcos.garcia@upm.es (M.G.A.)

**Keywords:** ultra-high performance concrete (UHPC), concrete, compressive strength, durability, nano additions, additions

## Abstract

This paper seeks to optimize the mechanical and durability properties of ultra-high performance concrete (UHPC). To meet this objective, concrete specimens were manufactured by using 1100 kg/m^3^ of binder, water/binder ratio 0.20, silica sand and last generation of superplasticizer. Silica fume, metakaolin and two types of nano silica were used for improving the performances of the concrete. Additional mixtures included 13 mm long OL steel fibers. Compressive strength, electrical resistivity, mercury intrusion porosimetry tests, and differential and thermogravimetric thermal analysis were carried out. The binary combination of nano silica and metakaolin, and the ternary combination of nano silica with metakaolin and silica fume, led to the best performances of the UHPC, both mechanical and durable performances.

## 1. Introduction

Research developed in recent decades has led significant advances in concrete technology that has enabled a continuous increase of the performances of concrete, mainly based on the improvement of the microstructure. The demand for concretes with improved performance changes the traditional approach for improving the mechanical properties of concrete. The terms such as high-strength concrete (HSC) and ultra-high strength concrete (UHSC) [1,2] are less used. In the early 1990s, Richard [3] introduced the term “reactive powder concrete” (RPC) to refer to low porosity concretes with high compression strength, which does not use coarse aggregates and also adds small particle materials such as silica fume. In the same decade, De Larrard and Naaman used the term ultra-high performance concrete (UHPC) to describe concrete of similar characteristics to that described by Richard, but also including the concept of high packing density [4,5,6].

Today, there is widespread consensus about the use of the acronym UHPC to refer to a cement-based composite material that has high compressive strength, above 150 MPa, which uses high amounts of cement, mineral additions with micrometric and nanometric particle sizes, fine aggregates with high SiO_2_ content, and very low water/binder ratio, even less than 0.20 (see ACI Committee 239 [7] and Federal Highway Administration, FHWA, [8]). The low water content requires the mandatory use of superplasticizer additives that allow for achieving better packaging of the particles of the composite material, and this provides greater fluidity and workability to the mixture [9]. This type of concrete, in addition to having a high compressive strength, provides other competitive advantages, among which stands out for its excellent durability. The increased strength of the concrete matrix is associated with the reduction of porosity, greater homogeneity and the improvement of its microstructure, and these in turn have a direct relationship to durability [5,10,11].

When HPC or UHPC incorporate structural fibers that can partially or completely replace traditional steel reinforcement, the terms high or ultra-high performance fiber reinforced concrete (HPFRC and UHPFRC) [12] can be used. Fibers improve the ductility and brittle nature of plain (fiberless) concrete under tensile stresses, as well as increase energy absorption capacity after the first fissure. In this type of concrete, high-pressure compaction and heat treatments are often applied to speed up the hydration process, reduce voids and increase density, leading to even higher compressive resistance [2,13]. The NF P18–710 standard [14] defines UHPFRC as a concrete with high compressive strength and high post-cracking tensile strength, giving it a ductile behavior in tension, whose lack of brittleness makes it possible to design and produce structures and structural members without using reinforced steel. While UHPC requires a very careful and tight mixing design, it can be obtained by modifying conventional concrete [15]. In this sense, the design of UHPC can be optimized through the use of artificial neural networks applied to an extensive experimental database [16,17]. Additionally, rheology-related aspects play an important role in the behavior of UHPC [18,19].

Pozzolanic materials, such as metakaolin, silica fume and nano silica are used as partial substitution of cement in mixing designs and are responsible for microstructure densification, porosity reduction and accelerating the clinker hydration process [20,21,22,23], resulting in improved concrete performance. The UHPC matrix densifies due to two types of phenomena: the pozzolanic reaction of these materials with calcium hydroxide, and the associated filler effect that is associated with the packaging density [24,25,26]. While it is true that both phenomena are inclusive of each other, it can be said that the filler effect usually occurs at an early age, while the pozzolanic reaction begins at a later age, the same one that is enhanced with pH and temperature. Not always maximum compactness leads to maximum compressive strength, with hydration processes playing an important role in time [25].

Firstly, hydration of the clinker minerals (alite C_3_S, belite C_2_S, tri-calcium aluminate C_3_A, and tetra-calcium ferro aluminate C_4_FA) occurs, forming phases of hydrated calcium silicate (C-S-H gel) and calcium hydroxide or portlandite (CH). The amount of CH produced after hydration is limited by UHPC’s own low water content and this in turn limits the pozzolanic reaction between portlandite and reactive particles from additions that produce other phases of C-S-H.

In addition, the use of particles with diameters of less order of magnitude than those used in conventional concretes improve the density of packaging. The use of fine aggregates gives greater uniformity to the matrix, while additions and nano additions are able to fill the gaps among cement grains and other materials constituting the matrix, even if their particles have not reacted with CH. Furthermore, on the surface of filler particles, there is a phenomenon of nucleation of hydrated calcium silicate (C-S-H); this phenomenon that accelerates cement hydration is known as ‘seeding effect’ [20,27] and directly influences the improvement of UHPC performance.

In order to maximize particle packaging, different models have been developed. In 1890, Feret proposed a law linking increased strength to increased compactness. In the 1930s, Andreasen and Andersen (A&A) [28] presented a semi-empirical study on packaging continuous particle size distributions (PSD) and then Funk and Dinger [29] made an adaptation of this model (modified A&A) considering the smaller particles that were supposed to be zero in the A&A model. De Larrard developed the Solid Suspension Model (SSM) based on the Linear Packaging Density Model (LPDM), developed by Money in 1951, but considering a non-infinite but elevated viscosity. All these models have been adapted to be able to integrate into different software that allow the optimization of the packaging of particles used in UHPC. Actual packaging will always be less than ideal packaging based on closed hexagonal packaging, so models consider adaptations that can approximate the actual state [4,6,30,31].

This study seeks to optimize the design of UHPC, combining high compressive strength with improved durability, through the use of additions and nano-additions. In addition to compressive strength tests, mercury intrusion porosimetry (MIP) and differential and thermogravimetric thermal analysis (ATD-TG) tests were performed. Such tests establish a correlation among the compressive strength of various UHPCs, with the modification of porous network, and the formation of the different hydrated products in the UHPC matrix. In addition, electrical resistivity tests were carried out to indirectly measure the connectivity and size of pores and relate them to the distribution obtained in the MIP tests.

## 2. Materials and Mix Design

For the different mixing dosages of UHPC, siliceous sand was used as the only aggregate, with a maximum particle size of 1 mm, specific gravity between 2400–2600 kg/m^3^ and SiO_2_ content above 98%. Portland cement CEM I 52.5R, according to EN 197-1 Standard [32], and produced by CEMEX Spain Operations, was used. This grey cement, composed mainly of clinker (95–100% by mass), is characterised by reaching a high compressive strength at early ages. Several types of additions and nano additions were employed in the different mixes produced: metakaolin (MK), silica fume (SF), and two types of nanosilica (NS_1_, NS_2_), named Aerosil OX-50 and Aerosil 200, respectively. MK is a highly reactive pozzolanic metakaolin, produced by thermal activation of high purity kaolin clay, composed of micron-sized particles, manufactured by SIKA under the name of Metamax. SF is composed by non-porous amorphous spheres of SiO_2_ with submicron sizes of approximately 0.15 microns and specific surface area of 15–30 m^2^/g, which tend to agglomerate, reaching particle sizes of up to 1 micron. It was manufactured by Elkem with the name of Elkem Microsilica 940 U. NS_1_ has an average particle size of 40 nanometers and a specific surface area of 35–65 m^2^/g. NS_2_ has an average particle size of 12 nanometers and a specific surface area of 175–225 m^2^/g. The granulometry of the additions can be seen together with the cement in Figure 1. Moreover, the *D_50_* particle size and the specific surface can be seen in Table 1.

The water used was from the drinking water supply system of Madrid (Spain). Due to the low water content in the mixes, a superplasticizer of the PCE type (polycarboxylate ether) was used, capable of dispersing the particles of materials and improving the workability of the mixtures. This product is a slightly viscous yellowish liquid, with a density of 1080 kg/m^3^, manufactured by Sika under the name of Sika Viscocrete 20-HE. Such product was used to achieve the spread diameter desired in the mixes. In the case of the fibre reinforced mixes, Dramix OL 13/0.20 steel fibres (13 mm in length and 0.20 mm in diameter) were also incorporated. These fibres are manufactured from cold drawn high tensile steel wire, and their dimensions and mechanical properties can be seen in Figure 2.

In previous studies [1,2], it was seen that a remarkable improvement of the mechanical properties were obtained with a 10% substitution of MK, showing also a refinement of the concrete pore structure. Regarding the content of silica fume, it was observed that a significant enhancement of the mechanical properties could be obtained, replacing 10% of cement with SF [3]. Similarly, other studies highlighted that an increment in the concrete durability could also be achieved by replacing 10% of cement weight by SF [4]. In what concerns the use of nano additions, there are contributions that have pointed out that an improvement of the mechanical properties and durability could be obtained by using different replacements of cement which range from low amounts close to 1% to up to 6% [5,6,9]. However, there are some publications that have studied the effect of the combination of ternary mixtures in the concrete durability when normal contents of cement are employed [10]. In almost all the aforementioned publications, though, the content of cements below 500 kg/m^3^ was used. Consequently, the effect of MK, SF and nano additions in UHPC mechanical properties and durability when used and combined in ternary, quaternary, and quinary mixtures is a matter that deserves to be studied.

Based on the previous studies mentioned, all the mixes performed were designed considering replacements around 10% of the cement weight. In all the mixtures, a Viscocrete 20 HE superplasticizer was added in the amount necessary to obtain the same spread diameter on the shaking table. Different combinations of MK, SF, and NS_1_ and NS_2_ were employed seeking not only an improvement of the mechanical properties, but also of the durability. The reference mixture, which is named Control (CTRL in Table 2), was the base of the rest of them. It was manufactured in order to compare the influence of additions, nano additions and the combination of nano additions with traditional additions used for improving mechanical properties and durability of UHPC mixes. For this blend, 1100 kg of cement per m^3^ of concrete, 1.60% of the cement weight of the superplasticizer additive, and a water–cement ratio of 0.20 was considered. The ratio volume of paste–volume/aggregate–volume was 1.27. The volume of paste represented 56% of the total volume of concrete, while the volume of aggregate, consisting only of silica sand, represented 44%.

Three binary mixes were carried out, two of them with 8% and 10% of MK, which were called 8 MK and 10 MK and another one manufactured with 8% of SF termed 8SF. Regarding the ternary mixes, two formulations were employed, the first one, where 10% of MK and 2% of Aerosil OX-50 were employed, which was named 10 MK + 2NS_1_, and the second one, where 4% of MK was mixed with 4% of SF, which was termed 4MK4SF. Moreover, two quaternary mixes were carried out. While one of them included 4% of MK, 4% of SF and 2% of Aerosil OX-50, which was called 4MK4SF + 2NS_1_, the other was termed 4MK4SF + 1.5NS_1_ and had a combination of 4% of MK, 4% of SF, and 1.5% of NS_1_. Lastly, a quinary mixture which was comprised of 4% of MK, 4% of SF, and 2% of a combination of Aerosil OX-50 and Aerosil 200 was named 4MK4SF + 2NS_C_. All the combinations performed can be seen in Table 2. In addition, to the mixes seen in Table 2, Dramix-OL fibres were added to the CTRL and 4MK4SF mixes in a 0.89% volumetric fraction. Such mixes were termed CTRL + fibres and 4MK4SF + fibres.

UHPC and UHPFRC mixtures were prepared using a two-speed automatic laboratory mixer following the procedure shown in Table 3.

It is worth noting that, when supplementary cementitious materials were used, they were mixed with the cement until reaching a homogeneous compound before starting the mixing. In the case of the mixes where fibres were added, they were incorporated in the fifth step. Once mixed, the fresh materials were poured into metal moulds in two layers of equal thickness, which were compacted after each pour with 120 strokes. Each mould allows for obtaining three prismatic specimens of dimensions 160 mm–40 mm–40 mm (length–width–height, respectively). The specimens were removed from the moulds after 24 h and kept in a climatic chamber (20 °C of temperature and at least 95% of humidity) until the test age.

## 3. Experimental Campaign

### 3.1. Fresh State

The determination of the consistency of the fresh UHPC was assessed, prior to the filling of the moulds, by means of the shaking table following the recommendation EN 1015-3 [33]. It should be mentioned that, in order to achieve similar fresh properties, the superplasticiser ranged from being 1.6% of the cementitious materials weight in the CTRL mix up to 3% in the 10 MK + 2NS_1_ formulation. Consequently, the slump flow of all mixes was 250 mm approximately. In Figure 3, an image of the mini-cone being filled and of the patty of one of the mixes can be seen.

### 3.2. Compressive Strength Test

The compressive strength of the mixes was assessed following EN 1015-11 [34] at 2, 7, and 28 days of age. In the case of CTRL + fibres and 4MK4SF + fibre mixes, tests were only conducted at 7 and 28 days of age. Prior to the compressive strength test, the specimens were tested using a three-point bending setup, and the resulting halves were employed in the compressive strength tests performed. An Ibertest hydraulic press, equipped with a load cell with a maximum capacity of 3000 kN, was used. The compressive strength tests were carried out by using a device that assured that the load exerted by the machine was applied on two metal plates of dimensions 40 × 40 mm^2^. As established in [34], the load rate was fixed at 2.4 kN/s. The testing machine, the loading device and the outlook of one of the samples with and without fibres after being tested can be seen in Figure 4.

### 3.3. Microstructural Characterization

In order to relate the compressive strength tests with the microstructure of the tested matrixes, differential thermal and thermogravimetric analysis (DTA-TG) and mercury intrusion porosimetry (MIP) tests were carried out at 28 days of age in the mixtures that did not include metallic fibres. At least two samples of each formulation were analysed in each test.

MIP test was carried out in cylindric samples extracted from undamaged samples by using a water-cooled column drill. The cylindrical specimens were placed for half an hour in a vacuum pump to extract the water present in the pores. They were then immersed in isopropanol for 24 h and dried in an oven at 40 °C until the time of the test. The cement hydration was stopped following the described procedure. The MIP analysis was performed on a Micromeritics Autopore IV 9500 equipment, operating up to a pressure of 33,000 psi (228 MPa) and covering a pore diameter range from 0.006 to 175 μm. Penetrometers of 5 cm^3^ of bulb and 0.366 cm^3^ of stem were used. The equipment used can be seen in Figure 5a.

In the case of the DTA-TG tests, the samples followed the same procedure as in the case of the MIP tests. Afterwards, they were manually grinded before using an automatic Retch RM200 mortar grinder, which reduced the size of the particles up to a point where the maximum particle was smaller than 0.5 mm. The dust obtained was stored in sealed plastic bags until the tests were performed. The DTA-TG tests were performed following an adaptation of ASTM E1131: 2008 [35] in a N_2_ environment using a simultaneous thermal analyser Setaram brand, model Labsys Evo, provided with a 0.1 mg precise balance. The procedure used a 10 °C/min heating ramp between 40 °C and 1100 °C and the crucibles used were made of alumina (α-Al_2_O_3_) previously calcined at 1500 °C, which was employed as reference material. The equipment used can be seen in Figure 5b.

The electrical resistivity test, which has the advantage of being a non-destructive test, was carried out according to UNE 83988-1 [36], using the Giatec RCON™ measuring equipment, capable of measuring the electrical resistance (*R_e_*) of concrete specimens using a uniaxial method within a frequency range of 1 Hz–30 kHz. The equipment applies a uniform electric field by means of two electrodes that contact the bases of the saturated concrete specimen through sponges previously moistened and saturated with gel. For each type of concrete, the average value of two measurements made at 28 days of curing was considered. Then, the electrical resistivity (*ρ_e_*) was calculated according to the following equation:(1)$$\rho_{e}=R_{e}k=R_{e}\frac{S}{L}$$
where *ρ_e_* is the electrical resistivity (Ω m), *k* is the cell constant (m), *R_e_* is the electrical resistance (Ω), *S* is the surface area of the specimen in contact with sponges (m^2^), and *L* is the height of the specimen.

## 4. Experimental Results

### 4.1. Compressive Strength

Table 4 shows the compressive tests results for all the mixes performed at 2, 7 and 28 days of age. Having the compressive strength values at different ages permitted not only comparing the strength of the mixes but also assessing the effect of the combination of nano additions and additions in the evolution of strength of the concrete mixes. In order to establish the possible experimental scattering, the coefficient of variation was also included.

In Table 4, it can be clearly perceived that the high amount of cement together with the reduced water to cement ratio generated a remarkable compressive strength in the CTRL formulation. It should be underlined that, even at two days of age, the compressive strength of the mentioned mix surpassed 100 MPa. Between two and 28 days of age, the compressive strength increased almost 25%, the development of strength being more relevant in the first week. In the case of the 8 MK mix, at two days, the compressive strength was also above 100 MPa. Moreover, the increment of strength between two and 28 days was close to 40%. Similarly as in the case of the CTRL mix, the compressive strength evolution was evident in the first week of age, but, at 28 days, there was a greater increment if such value is compared with the one at two days. The compressive strength of the 10 MK mix was slightly below 100 MPa at two days of age. Although a noticeable increment of resistance was noticed during the first week of age of the samples, the enhancement of mechanical properties was greater between seven and 28 days. In fact, a 10 MK mix virtually reached the same strength than the 8 MK mix at 28 days even though the initial values registered were smaller. A similar trend can be detected when analysing the results of the 10 MK + 2NS_1_ mix. A two-day strength was registered below 100 MPa followed by a rising of the resistance as time passed until 135 MPa were reached at 28 days of age. No improvement of the compressive strength was detected when NS_1_ was combined MK. The evolution of resistance of the 8SF mix is quite similar to the one detected in the 8 MK mix. In the case of the 4MK4SF formulation, the initial compressive strength was above 100 MPa, being the greatest of all the mixes tested. Such strength evolved up to values close to the ones of the previously commented mixes not being the highest at 28 days of age. When 2% of NS_1_ was added to 4% of MK and 4% of SF, a remarkable improvement of the strength could be noticed between seven and 28 days of age reaching over 140 MPa. It is curious that, if NS_1_ is reduced to 1.5%, the compressive strength of the material is almost stable, being close to the highest value registered (4MK4SF + 1.5NS_1_ mix). Lastly, 4MK4SF + 2NS_C_ shows similar trends to the ones that can be perceived for most of the mixes.

In contrast to the homogeneity of the compressive strength values previously mentioned, the mixes where the steel fibres were added showed a notable increment of compressive strength between seven and 28 days of age. In addition, it should be underlined that the 4MK4SF + fibres mix reached 171 MPa of compressive strength at 28 days of age. There is not a complete assessment of the strength evolution in these mixes due to the absence of test results at two days of age. In any case, it should be highlighted that, in all tests and at all ages, the experimental scatter was remarkably low due to the careful manufacturing, curing, and testing that was carried out.

Figure 6 shows the comparison of the compressive strength of the mixes considering that the CTRL mix strength is 1. In such figure, it can be perceived that all the mixes where a certain proportion of cement was substituted by a pozzolanic addition showed a greater compressive strength. The increment of the compressive strength of the mixes if compared with CTRL were between eight and 16% at 28 days of age. However, it should also be mentioned that the differences that appear among the mixes with additions and without fibres are scarce and lay within the coefficient of variation values. This aspect is valid independently of the age of the mix. Based on the previous comments, it might be stated that there is no clear influence of the additions and their combinations in the compressive strength of the mixes analysed. On the contrary, when steel fibres are added to the CTRL (CTRL + fibres) and the 4MK4SF (4MK4SF + fibres) mixes, the presence of fibres clearly improved the behaviour of the correspondent mixes, reaching up to 30.4% and 37.9% increment at 7 and 28 days, respectively, compared with the control mixture without fibres.

### 4.2. Mercury Intrusion Porosimetry (MIP)

The results of the MIP test can be seen in Figure 7. It should be mentioned that MIP tests were only carried out in the mixes without fibres. While, in Figure 7a, the results of the logarithm of differential intrusion versus the pore size appear, in Figure 7b, the cumulative intrusion versus the pore size appear. The first characteristic that can be perceived in Figure 7a is that CTRL boasts a noticeable higher pore critical diameter if compared with the mixes where additions were used. Moreover, the area below the curve of the CTRL mix is remarkably greater than in the rest of the cases (see Figure 7b). If a more detailed analysis of the test results is performed, it can be clearly seen that, for the mixes that combined 4% of MK and 4% of SF, a certain content of nano additions boasted a reduced size of the pore critical diameter. Moreover, the area below such curves was also noticeably smaller than the correspondent of the rest of the mixes. Analysing such curves and comparing them with the one of CTRL, it could be clearly seen that the pore critical diameter has clearly shifted towards the left, which means a smaller size of the pore critical diameter. A similar, but at the same time more subtle effect, can be seen if the CTRL curve that is compared with the ones of the mixes was with traditional additions (MK and SF), or if even a combination of them was employed. In such cases, the reduction of the size of the pore critical diameter is less evident. Similarly, the area below the curves is smaller than in the case of CTRL but at the same time greater than the one of the mixes that included a combination of 4% MK, 4% SF and a certain content of nano additions. It should be also highlighted that the critical pore size of the mixes where only MK was used boasted a greater pore critical diameter than those where SF were employed.

If the cumulative intrusion registered in the tested mixes is compared, it can be clearly seen that similar trends to the ones previously mentioned appear. The CTRL formulation registers the greatest intrusion of mercury. On the contrary, the mixes 4MK4SF + 1.5NS_1_, 4MK4SF + 2NS_1_ and 4MK4SF + 2NS_C_ not only were the ones where a more limited amount of mercury intruded, but they also boasted the smallest pore threshold diameter. The results in these three formulations were remarkably similar. The mixes in which MK and SF were used showed a smaller amount of mercury intruded if compared with the CTRL mix. However, if the effect of both additions is compared, it can be seen that SF was not equally effective compared to MK. The use of SF generates a greater reduction of the amount of intruded mercury than MK. Oppositely, there seems to be no remarkable differences in the pore threshold diameter of the mixes independently of the use of SF or MK.

If the values of total porosity, median pore diameter, threshold diameter, total intrusion volume and tortuosity are extracted from the test results, Table 5 can be performed. In this table, it can be seen that the presence of nano additions, when combined with 4% of MK and 4% of SF, reduced by up to 34% the porosity of the mixes. It is also true that SF and MK by themselves are capable of lessening the porosity, although in slightly lower percentages than those obtained with nano additions. Regarding the variation of the pore threshold diameter, it could be said that, when SF and MK are added independently, there is no clear trend; however, this parameter is reduced with the presence of nano additions. Where the median pore diameter is concerned, the use of nano additions reduces its size more than 50% while the use of traditional additions only reaches reductions of around 30%.

### 4.3. Differential Thermal and Thermogravimetric Analysis (DTA-TG)

In addition, the mixes were analyzed by Differential Thermal Analysis (TGA/DTA). TGA/DTA profiles, according to ASTM E 1131 [35]. Plotting the derivative of the weight loss or what is the same, the speed of weight loss (dTG) vs. temperature is more useful than the representation of the weight loss (TG) vs. temperature as it allows a more clear and unequivocal identification of the different starting and ending temperatures of different processes of weight loss with temperature. From the temperatures selected for the representation of dTG vs. TG, the quantification of the weight loss with the temperature associated with the different reactions that occur in the cement was carried out. Figure 8 shows the dTG vs. temperature of the mixes. Based on the methods proposed by Bhatty [37], Pane et al. [38] and Monteagudo et al. [39], various regions of dehydration, *Ldh* (region *T*_1_), dehydroxylation, *Ldx* (region *T*_2_) and decarbonation, *Ldc* (region *T*_3_), are highlighted in Figure 8.

Table 6 shows the Portlandite, C-S-H gel and Portlandite (CH) contents based on the TG results at the age of 28 days for all the mixtures. The key values for comparing the influence of the additions and nanoadditions in the cement hydration are the total combined water (*T*_1_ + *T*_2_ + 0.41 × *T*_3_), the equivalent calcium content or equivalent Portlandite (CH) ((*T*_2_ + 0.41 × *T*_3_) and the ratio water of C-S-H gel/Portlandite (CH) (*T*_1_/(*T*_2_ + 0.41 × *T*_3_).

The ratio water of C-S-H gel/CH increases when the cement additions are incorporated to the mixes, being the highest values shown by the mixtures 10 MK + 2NS_1_ and 4MK4SF + 1.5NS_1_, both including nano silica. In addition, the value of the CH is the lowest for both mixtures. This is consistent with the high degree of hydration that nano silica usually produces at the age of 28 days. The rest of mixtures show lower hydration values compared with those showing 10 MK + 2NS_1_ and 4MK4SF + 1.5NS_1_. The lowest value of the ratio water of C-S-H gel/CH is shown by the CTRL mixture.

### 4.4. Electrical Resistivity

The electrical resistivity in a saturated concrete is an indirect measure of the connectivity and the size of its pores. This volumetric property, according to Ohm’s Law, relates the resistance of the material to the passage of electric charges with a geometric factor that is a function of the length and cross section of the specimen. These tests were carried out according to the UNE 83988-1 [36]. In Table 7, it can be seen that all the mixes in which additions were used registered resistivity values one order of magnitude higher than the control formulation. Moreover, if no nano additions are used, SF seems to be more apt for reducing the connected porosity than MK. In the cases where nanoadditions were employed, a notable increment of the resistivity found can be perceived. In addition, no notable effects of NS_2_ were detected. It should be highlighted that the highest value of resistivity was found in the mix with the greatest substitution of cement by additions.

## 5. Discussion

Figure 9 and Figure 10 show the total porosity of the samples at 28 days, and the distribution of the pores according to their size, expressed as percentages of the total porosity, respectively. These graphs were elaborated from the cumulative mercury intrusion curves shown in Figure 7b, where the total porosity equivalent to 100% of the volume of mercury intrusion is shown.

As shown in Figure 9, samples with additions reveal lower total porosity at 28 days compared with the control sample, which already has low porosity. A minimum porosity percentage of 5.91% was reached in the case of the 4MK4SF + 2NS_1_ sample, which represents a porosity reduction of 34%. This increase in the density of the UHPC matrices is coherent with a higher value of hydration shown by the ratio water of C-S-H gel/CH and the equivalent calcium content in DTA-TG analysis. In particular, the highest values of the ratio water of C-S-H gel/CH are reached by 4MK4SF + 1.5NS_1_ and 10 MK + 2NS_1_, which correspond with the lowest values of the total porosity. In addition, both mixtures show a high resistivity value and the best compressive strength (see Table 4).

The percentual distribution of the porous network is presented in Figure 10. It shows that, in all cases, the medium size capillaries predominate, with diameters between 10 ηm and 50 ηm. Large capillaries, with diameters between 50 ηm and 10 μm, seem to decrease with the use of additions, while the percentage of small capillaries, with diameters between 5 and 10 ηm, increases. The increase in the percentage of small capillaries is very evident with the use of nanosilica. Figure 10 shows, for example, the mixtures 10 MK and 10 MK + 2NS_1_, whose percentages of small capillaries are 5.6% and 7.2%, respectively, which represents an increase of 28.6% when replacing 2% of the cement content with nanosilice OX-50. Similarly, the values of 4MK4SF and 4MK4SF + 2NS_1_ can be observed, with percentages of small capillaries of 5.4% and 6.4%, respectively, which represents an increase of 18.5% in the percentage of these capillaries when replacing 2% of the cement content per nanosilice OX-50. In both cases, the total porosity is also reduced, as shown in Figure 9.

In addition, Figure 11 shows the relation between the ratio water of C-S-H gel/CH and the compressive strength of the different specimens.

The use of additions such as metakaolin and silica fume increases compressive strength significantly as shown in Table 4, in addition to reducing the total porosity and especially the large capillaries as seen in Figure 9 and Figure 10, respectively. When nanosilica is added to the pozzolanic materials mentioned above, the compressive strength is not greatly affected, but, in contrast, the densification of the matrix and the corresponding reduction in porosity are significantly favored.

In Figure 12, it can be seen how the use of additions, especially nanosilica, is reflected in a very significant increase in resistivity compared with the control sample. For example, in the 10 MK and 10 MK + 2NS_1_ mixtures, a slight increase in the compressive strength is observed when 2% of the weight of the cement is substituted by nanosilice OX-50, while its resistivity practically doubles. In the case of mixtures containing 4% of metakaolin and 4% of silica fume, it can be seen how the use of nanosilica slightly increases the compressive strength in the mixtures 4MK4SF + 2NS_C_ and 4MK4SF + 1.5NS_1_, and in a higher grade in 4MK4SF + 2NS_1_. In all these last cases, the resistivity increases significantly, between 45% and 54% with respect to the sample 4MK4SF, which supposes a greater resistance to the penetration of chlorides and CO_2_ of this type of UHPC, and this translates into a greater durability.

## 6. Conclusions

The addition of silica fume and metakaolin to the Portland cement CEM I 52.5R, separately or combined, in total percentages of 8%, improves the compressive strength of the concrete by about 10–12%. If these micro additions are combined with small amounts of nano silica, on the order of 1.5 to 2%, the compressive strength improves up to 16%. Being a significant improvement, the greatest contribution of this combination of additions (silica fume and metakaolin) with nanosilica lies in the improvement of the microstructure of the hydrated products and the significant reduction of porosity.

This is explained both by the pozzolanic reaction of these materials and by the filler effect associated with the size of its particles, as was previously observed with other combinations of nano additions with cement [25].

The use of additions and nanoaditions increases the formation of CSH gel while decreasing the amount of total portlandite during the hydration process of the UHPC matrix. As is known, this shows the contribution of micro and nano additions to the formation of secondary CSH gel, which results in the improvement of the performance of hydrated cement products.

The use of nanosilica plays a significant role in reducing the percentage of large capillaries and increasing the percentage of small capillaries, which translates into high resistivity values associated with greater durability of the UHPC matrix.

The combined use of silica fume and metakaolin with nanosilica improves the electrical resistivity up to 54% compared with using silica fume and metakaolin without nanosilica. In addition, the increase of the small capillaries of the porous network is increased by 18.4%, which indicates that the porous network is refined, also increasing the parameter of tortuosity.

For all of the above, the fundamental contribution of the studied combinations of silica fume, metakaolin and nanosilica lies in the improvement of the durable performance of concrete.

## Figures and Tables

**Figure 1 materials-14-06944-f001:**
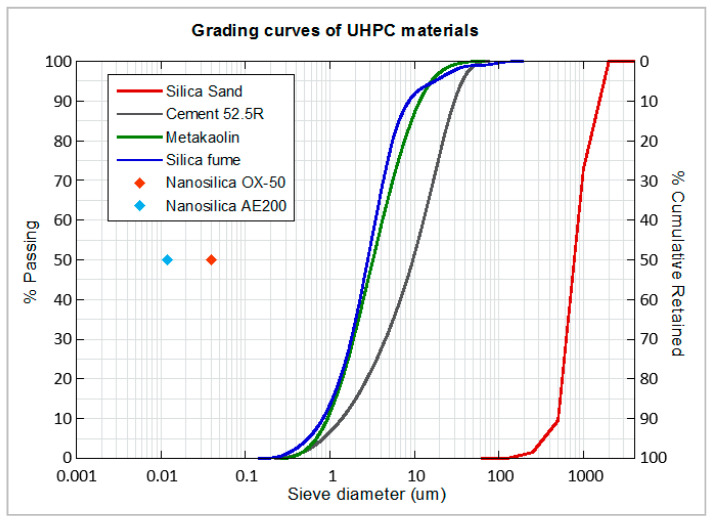
Grading curves of raw materials.

**Figure 2 materials-14-06944-f002:**
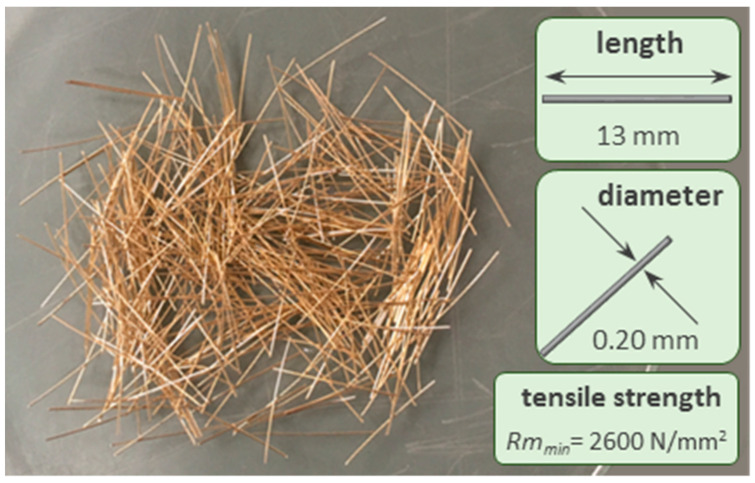
Grading Dramix-OL steel fiber characteristics.

**Figure 3 materials-14-06944-f003:**
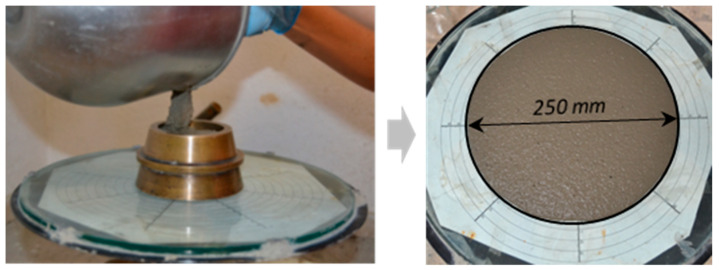
Determination of the consistency of fresh UHPC by shaking table. Spread diameter.

**Figure 4 materials-14-06944-f004:**
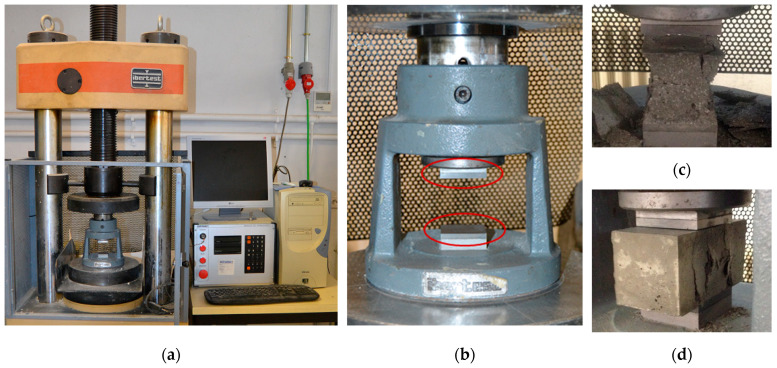
Compressive strength test: (**a**) hydraulic press; (**b**) device with steel supports of 40 mm × 40 mm; (**c**) test specimen fracture without fibres; (**d**) test specimen fracture with fibres.

**Figure 5 materials-14-06944-f005:**
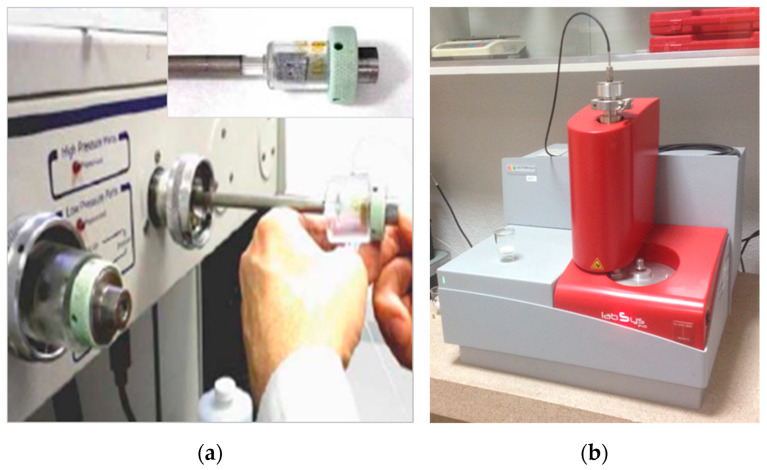
(**a**) Equipment used for MIP tests and penetrometer detail with cylindrical sample inside; (**b**) simultaneous thermal analyser used for DTA-TG tests.

**Figure 6 materials-14-06944-f006:**
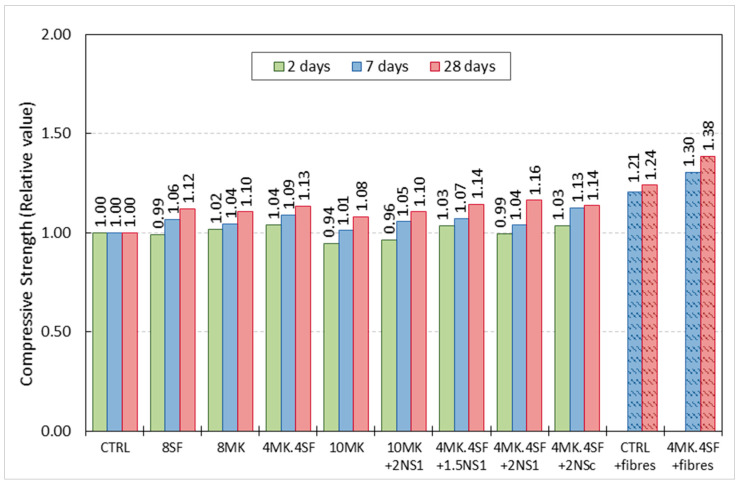
Relative increment of strength. CTRL mix results are taken as the unity at two, seven and 28 days of age for reference.

**Figure 7 materials-14-06944-f007:**
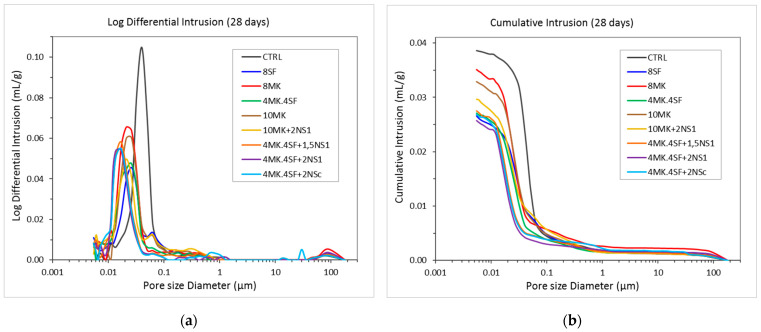
Graphs of MIP tests at 28 days: (**a**) Logarithm differential of mercury intrusion; (**b**) cumulative intrusion volume.

**Figure 8 materials-14-06944-f008:**
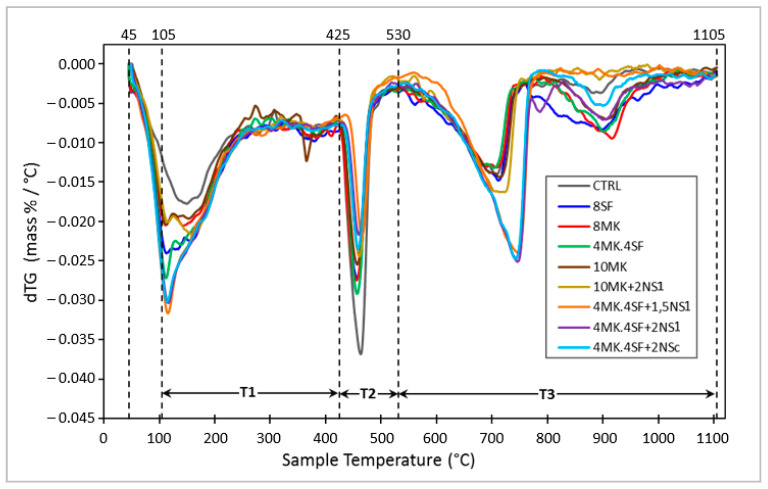
Curves of the thermogravimetric derivative according to temperature ranges.

**Figure 9 materials-14-06944-f009:**
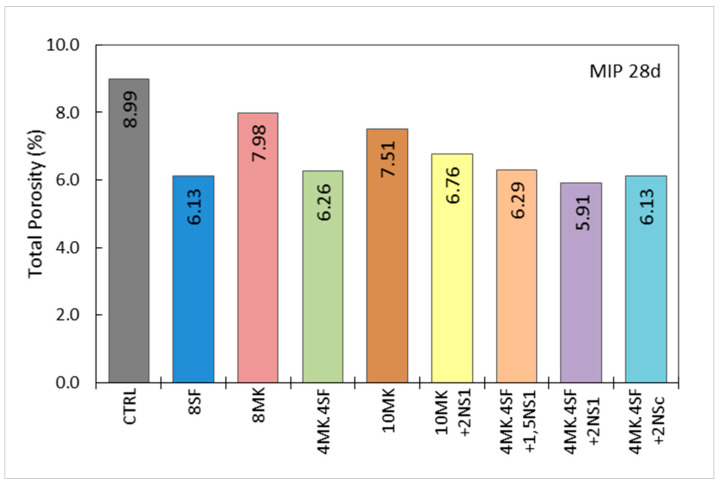
Curves of the thermogravimetric derivative according to temperature ranges.

**Figure 10 materials-14-06944-f010:**
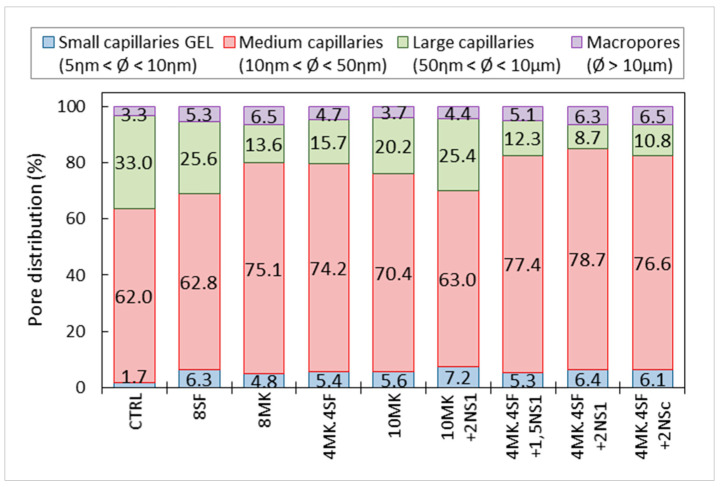
Curves of the thermogravimetric derivative according to temperature ranges.

**Figure 11 materials-14-06944-f011:**
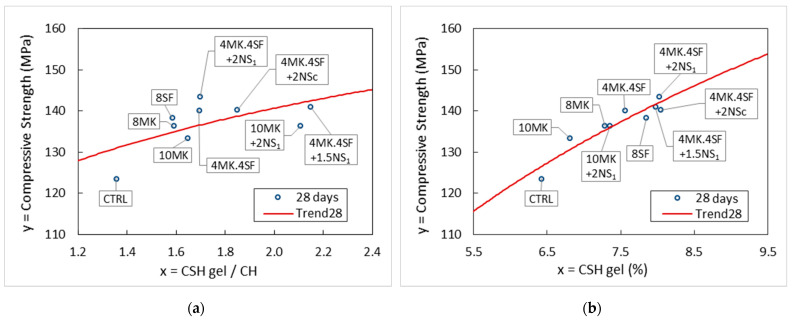
Influence of portlandite (**a**) and CSH gel (**b**) on compressive strength at 28 days.

**Figure 12 materials-14-06944-f012:**
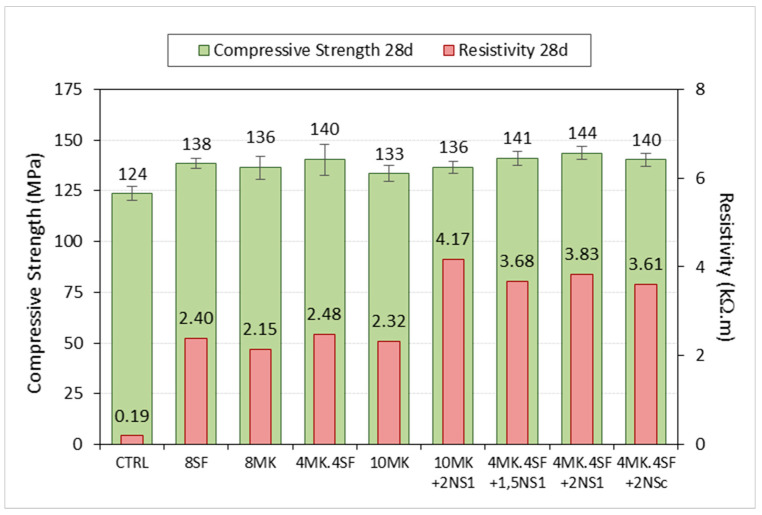
Influence of additions and nano additions on resistivity and compressive strength.

**Table 1 materials-14-06944-t001:** *D_50_* particle size and specific surface of raw materials.

Material	Particle Size *D_50_*		Specific Surface Area (m^2^/g)
Silica sand (0.7 mm)	0.750	mm	-
Cement (type I–52.5R)	9.630	µm	1.31
Metakaolin (Metamax)	3.240	µm	12.56
Silica fume (Norway SF)	2.820	µm	21.59
Nano silica (Aerosil OX-50)	0.040	µm	50 ± 15
Nano silica (Aerosil 200)	0.012	µm	200 ± 25

**Table 2 materials-14-06944-t002:** Proportioning of UHPC mixes.

Material	Units	CTRL	8SF	8 MK	4MK4SF	10 MK	10 MK + 2NS_1_	4MK4SF + 1.5NS_1_	4MK4SF + 2NS_1_	4MK4SF + 2NS_C_
Cement	[kg/m^3^]	1100	1012	1012	1012	990	968	995.5	990	990
Silica fume (SF)	[kg/m^3^]	0	88	0	44	0	0	44	44	44
Metakaolin (MK)	[kg/m^3^]	0	0	88	44	110	110	44	44	44
Nanosilica OX-50 (NS_1_)	[kg/m^3^]	0	0	0	0	0	22	16.5	22	16.5
Nanosilica AE-200 (NS_2_)	[kg/m^3^]	0	0	0	0	0	0	0	0	5.5
Water	[kg/m^3^]	220	220	220	220	220	220	220	220	220
Silica sand	[kg/m^3^]	1211	1211	1211	1211	1211	1211	1211	1211	1211
Superplastizicer (SP)	[kg/m^3^]	17.6	25.3	26.4	23.1	26.4	33.0	26.4	27.5	30.8
(% of cement + additions)		1.6%	2.3%	2.4%	2.1%	2.4%	3.0%	2.4%	2.5%	2.8%
Total	[kg/m^3^]	2548.6	2556.3	2557.4	2554.1	2557.4	2564.0	2557.4	2558.5	2561.8
V*_paste_*/V*_aggregate_*		1.28	1.33	1.32	1.32	1.32	1.33	1.32	1.33	1.33
Spread diameter	[mm]	250	250	250	250	250	250	250	250	250

**Table 3 materials-14-06944-t003:** Mixing sequence.

Steps	Time (s)
1. Addition of 75% of water and all the cementitious materials.	-
2. Low speed mixing.	30
3. Low speed mixing while sand is added.	30
4. High speed mixing.	30
5. Resting time. Addition of 25% of water + SP (20 s before the end of this period)	90
6. High speed mixing.	60

**Table 4 materials-14-06944-t004:** Compressive strength values of UHPC studied at different ages.

Compressive Strength	CTRL	8SF	8 MK	4MK4SF	10 MK	10 MK + 2NS_1_	4MK4SF + 1.5NS_1_	4MK4SF + 2NS_1_	4MK4SF + 2NS_C_	CTRL + Fibres	4MK4SF + Fibres
*f_ck,2d_* [MPa]	102	101	103	106	96	98	105	101	105		
(%CV)	(2.9)	(4.9)	(0.5)	(0.4)	(2.5)	(1.9)	(2.6)	(4.6)	(0.5)		
*f_ck,7d_* [MPa]	112	119	117	122	113	118	120	116	126	135	146
(%CV)	(1.1)	(0.6)	(0.1)	(7.6)	(2.1)	(3.0)	(0.8)	(6.7)	(3.6)	(1.9)	(1.7)
*f_ck,28d_* [MPa]	124	138	136	140	133	136	141	144	140	153	171
(%CV)	(2.8)	(1.8)	(4.1)	(5.3)	(3.0)	(2.2)	(2.5)	(2.2)	(2.3)	(2.0)	(2.6)

**Table 5 materials-14-06944-t005:** Parameters of the mercury intrusion porosimetry tests at 28 days.

Parameters	Units	CTRL	8SF	8 MK	4MK4SF	10 MK	10 MK + 2NS_1_	4MK4SF + 1.5NS_1_	4MK4SF + 2NS_1_	4MK4SF + 2NS_C_
Total Intrusion Volume	[mL/g]	0.0386	0.0265	0.0351	0.0271	0.0329	0.0297	0.0275	0.0258	0.0268
Median Pore Diameter	[ηm]	44.3	31.6	27.1	27.1	28.7	26.9	20.4	19.2	20.0
Threshold Diameter	[ηm]	77.1	95.3	50.4	50.4	77.1	77.1	40.3	40.3	40.3
Porosity	[%]	8.99	6.13	7.98	6.26	7.51	6.76	6.29	5.91	6.13
Tortuosity		4.45	4.58	4.56	4.23	4.21	10.37	9.84	4.84	3.87

**Table 6 materials-14-06944-t006:** Percentage of water loss according to temperature ranges at 28 days.

Parameters	Units	CTRL	8SF	8 MK	4MK4SF	10 MK	10 MK + 2NS_1_	4MK4SF + 1.5NS_1_	4MK4SF + 2NS_1_	4MK4SF + 2NS_C_
*T*_1_ (105–425 °C)	[%]	6.43	7.85	7.28	7.56	6.81	7.35	7.97	8.02	8.04
*T*_2_ (425–530 °C)	[%]	2.94	2.26	2.27	2.25	2.04	1.79	1.60	1.74	1.82
*T*_3_ (530–1105 °C)	[%]	4.39	6.57	5.65	5.39	5.11	4.16	5.16	7.29	6.18
Total combined water (*T*_1_ + *T*_2_ + 0.41 × *T*_3_)	[%]	11.17	12.80	11.87	12.02	10.94	10.85	11.68	12.75	12.39
Equivalent calcium (CH) (*T*_2_ + 0.41 × *T*_3_)	[%]	4.74	4.96	4.58	4.46	4.13	3.49	3.71	4.73	4.35
CSH gel/CH *T*_1_/(*T*_2_ + 0.41 × *T*_3_)		1.36	1.58	1.59	1.69	1.65	2.11	2.15	1.70	1.85

**Table 7 materials-14-06944-t007:** Resistivity values of the main mixtures.

Parameters	Units	CTRL	8SF	8 MK	4MK4SF	10 MK	10 MK + 2NS_1_	4MK4SF + 1.5NS_1_	4MK4SF + 2NS_1_	4MK4SF + 2NS_C_
Resistivity	[kΩ.m]	0.184	2.400	2.150	2.485	2.325	4.170	3.680	3.830	3.610

## Data Availability

The data presented in this study are available on request from the corresponding author.

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
