# Peer review of "Matrix Optimization of Ultra High Performance Concrete for Improving Strength and Durability"

_materials, 2021, doi:10.3390/ma14226944_

Round 1
Reviewer 1 Report
The submitted manuscript presents interesting research in the field of Matrix optimization of Ultra High Performance Concrete for improving strength and durability.
Potentially, the article is an interesting topic for the readers of the journal Materials.
Some information from the solved area is known.
The informative value of the manuscript and the presentation of new knowledge must be improved overall.
Weaknesses of the research include the limited scope of the experimental program. Important mechanical properties of concrete also include tensile strength of concrete, modulus of elasticity, long-term shrinkage tests and others.
It would be appropriate to extend the experimental program (recommendations only).
It would be appropriate to extend the solved area by the area of electron microscopy or testing of structural elements (recommendations only).
The topic itself is solved logically.
The manuscript has the usual structure.
The description of the research and content of the manuscript itself can be understood.
Extensive research is underway in the area of experiments, analysis, and diagnostic of High Performance Concrete, when it is necessary to
rework and expand the information in the introduction section.
Interesting research about High Performance Concrete can be found in: https://doi.org/10.3390/cryst11040427 ; https://doi.org/10.3390/ma14195707 ; https://doi.org/10.3390/ma14143982
Correct the numbering of the images.
Figure 1 is twice. page 3 and 4.
Figure 1. Dramix-OL steel fibers characteristics - is blurred.
Table 3 table is very wide.
line 237 Table 4 - Wrong numbering of tables is given
Table 5 is missing.
References are not in the correct format. Fix it.
Very little relevant information and references are given. Add at least 6 references.
The manuscript must be prepared with greater care according to the MDPI template.
Improve the label and informative value - Figure 4. a) Simultaneous t .......
Indicate the standard deviations or the coefficient of variation for the test results. (if possible)
Figure 8. Divide the picture into two separate Figures.
Enlarge the figures. The text is small.
It is also necessary to improve the discussion on results and new knowledge. Also, focus on the conclusion and introduction section.
The manuscript must be revised.
Author Response
Comments and Suggestions for Authors
The submitted manuscript presents interesting research in the field of Matrix optimization of Ultra High Performance Concrete for improving strength and durability.
Potentially, the article is an interesting topic for the readers of the journal Materials.
Thank you for the kind comment.
Some information from the solved area is known.
The informative value of the manuscript and the presentation of new knowledge must be improved overall.
Weaknesses of the research include the limited scope of the experimental program. Important mechanical properties of concrete also include tensile strength of concrete, modulus of elasticity, long-term shrinkage tests and others.
It would be appropriate to extend the experimental program (recommendations only).
It would be appropriate to extend the solved area by the area of electron microscopy or testing of structural elements (recommendations only).
The topic itself is solved logically.
The manuscript has the usual structure.
The description of the research and content of the manuscript itself can be understood.
The authors agree with the reviewer comments. Nevertheless, the work was focused on studying the improvement of compressive strength and durability-related aspects of microstructure when micro additions (silica fume and metakaolin) and nano silica are combined in UHPC.
The analysis of the hydration products by DTA-TG, the results of the porosimetry and electrical resistivity have allowed to relate the mechanical properties (compressive strength) with the modification of the microstructure and its influence on durability. The authors understand that the extensive experimental campaign, with a large number of combinations, eleven in total, and tests, has allowed this objective to be covered. It would be the subject of another paper to analyze in detail the rest of the mechanical properties of the UHPC, such as tensile strength, modulus of elasticity and long-term shrinkage, among others.
Extensive research is underway in the area of experiments, analysis, and diagnostic of High Performance Concrete, when it is necessary to
rework and expand the information in the introduction section.
Interesting research about High Performance Concrete can be found in: https://doi.org/10.3390/cryst11040427 ; https://doi.org/10.3390/ma14195707 ; https://doi.org/10.3390/ma14143982
These references have been included.
Correct the numbering of the images.
Figure 1 is twice. page 3 and 4.
Figure 1. Dramix-OL steel fibers characteristics - is blurred.
Table 3 table is very wide.
line 237 Table 4 - Wrong numbering of tables is given
Table 5 is missing.
References are not in the correct format. Fix it.
All these issues have been corrected.
Very little relevant information and references are given. Add at least 6 references.
Fourteen new references have been included.
The manuscript must be prepared with greater care according to the MDPI template.
Improve the label and informative value - Figure 4. a) Simultaneous t .......
All these issues have been corrected.
Indicate the standard deviations or the coefficient of variation for the test results. (if possible)
The coefficients of variation has been included in Table 4.
Figure 8. Divide the picture into two separate Figures.
Corrected
Enlarge the figures. The text is small.
Done
It is also necessary to improve the discussion on results and new knowledge. Also, focus on the conclusion and introduction section.
The manuscript must be revised.
The introduction and the conclusion section have been updated
The authors really appreciate the reviewer comments that improve the quality of the manuscript
Reviewer 2 Report
The paper presents the results of an interesting experimental campaign. The reviewer only argues that the conclusions section is a bit too hasty. The concise presentation is certainly an added value in a conclusions section. However, it should be briefly presented not only the results but also how these results were obtained. Therefore, the reviewer recommends making this section more comprehensive.
In addition to this, many problems with the numbering of the figures and tables must be reported. The reviewer does not exclude that the numbering errors may be the result of the conversion into pdf format and not actually present in the original file. However, the reviewer saw fit to list them below.
Line 121
“Figure 1. Dramix-OL steel fibres characteristics.”
Check the caption of the figure. This is Figure 2, not Figure 1.
Lines 156-157
“All the combinations performed can be seen in Table 1. In addition, to the mixes seen in Table 1, …”
The Authors probably refer to Table 2, not Table 1.
Line 160
“Table 1. Proportioning of UHPC mixes”
Check the caption of the table. This is Table 2, not Table 1.
Lines 162-163
“UHPC and UHPFRC mixtures were prepared using a two-speed automatic laboratory mixer following the procedure shown in Table 2.”
The Authors probably refer to Table 3, not Table 2.
Line 164
“Table 2. Mixing sequence.”
Check the caption of the table. This is Table 3, not Table 2.
Lines 180-181
“In Figure 2 an image of the mini-cone being filled and of the patty of one of the mixes can be seen.”
The Authors probably refer to Figure 3, not Figure 2.
Line 183
“Figure 2. Determination of the consistency of fresh UHPC by shaking table. Spread diameter.”
Check the caption of the figure. This is Figure 3, not Figure 2.
Lines 196-197
“Figure 3. Compressive strength test: a) hydraulic press, b) device with steel supports of 40mm x 40mm, c) test specimen fracture without fibres, d) test specimen fracture with fibres.”
Check the caption of the figure. This is Figure 4, not Figure 3.
Lines 233-234
“Figure 4. a) Simultaneous thermal analyser used for TDTA-TG tests. b) Equipment used for MIP tests and penetrometer detail with cylindrical sample inside.”
Check the caption of the figure. This is Figure 5, not Figure 4.
Line 242
“Table 3. Compressive strength values of UHPC studied at different ages.”
Check the caption of the table. This is Table 4, not Table 3.
Lines 289-290
“Figure 5. Relative increment of strength. CONTROL mix results are taken as the unity at two, seven and 28 days of age for reference.”
Check the caption of the figure. This is Figure 6, not Figure 5.
Lines 293-295
“While in Figure 7a appear the results of the logarithm of differential intrusion versus the pore size, in Figure 7b appear the cumulative intrusion versus the pore size.”
Check the order of the words.
Lines 323-324
“Figure 6. Graphs of MIP tests at 28 days. a) Logarithm differential of mercury intrusion. b) Cumulative intrusion volume.”
Check the caption of the figure. This is Figure 7, not Figure 6.
Line 334
“Table 4. Parameters of the mercury intrusion porosimetry tests at 28 days.”
Check the caption of the table. This is Table 5, not Table 4.
Line 347
“Figure 7. Curves of the thermogravimetric derivative according to temperature ranges.”
Check the caption of the figure. This is Figure 8, not Figure 7.
Line 377
“Table 5. Resistivity values of the main mixtures.”
Check the caption of the table. This is Table 7, not Table 5.
Line 380
“Figure 8 shows the total porosity of the samples at 28 days”
The Authors probably refer to Figure 9, not Figure 8.
Lines 381-382
“These graphs were elaborated from the cumulative mercury intrusion curves shown in Figure 6b”
The Authors probably refer to Figure 7b, not Figure 6b.
Line 388
“Figure 8. a) Total porosity at 28 days. b) Distribution of the porous network at 28 days.”
Check the caption of the figure. This is Figure 9, not Figure 8.
Lines 389-390
“As evidenced in Figure 8a, samples with additions reveal a lower total porosity at 28 days compared with the control sample, …”
The Authors probably refer to Figure 9a, not Figure 8a.
Lines 396-397
“In addition, both mixtures show a high resistivity value and the best compressive strength (see Table 3).”
The Authors probably refer to Table 4, not Table 3.
Line 398, 402-403
“The distribution of the porous network, which is presented in Figure 8b.”
“Figure 8b shows, for example, the mixtures 10MK and 10MK+2NS1,”
The Authors probably refer to Figure 9b, not Figure 8b.
Line 408
“In both cases, the total porosity is also reduced, as shown in Figure 8a.”
The Authors probably refer to Figure 9a, not Figure 8a.
Line 409-410
“In addition, Figure 9 shows the relation between the ratio water of C-S-H gel/cement content and the compressive strength of the different specimens.”
The Authors probably refer to Figure 10, not Figure 9.
Line 413
“Figure 9. Influence of portlandite and CSH gel on compressive strength at 28 days.”
Check the caption of the figure. This is Figure 10, not Figure 9.
Lines 414-415
“The use of additions such as metakaolin and silica fume increase compressive strength significantly as shown in Table 3,”
The Authors probably refer to Table 4, not Table 3.
Line 416
“as seen in Figure 8”
The Authors probably refer to Figure 9, not Figure 8.
Line 421
“Figure 8. Influence of additions and nanoaditions on resistivity and compressive strength .”
Check the caption of the figure. This is Figure 11, not Figure 8.
Author Response
Comments and Suggestions for Authors
The paper presents the results of an interesting experimental campaign. The reviewer only argues that the conclusions section is a bit too hasty. The concise presentation is certainly an added value in a conclusions section. However, it should be briefly presented not only the results but also how these results were obtained. Therefore, the reviewer recommends making this section more comprehensive.
Thank you for the kind comments
The conclusions section has been extended to
The authors agree with the reviewer comments.
The introduction and the conclusion sections have been extended and 14 new references added.
In addition to this, many problems with the numbering of the figures and tables must be reported. The reviewer does not exclude that the numbering errors may be the result of the conversion into pdf format and not actually present in the original file. However, the reviewer saw fit to list them below.
All the issues with the numbering and format of the Figures and Tables have been solved.
The authors really appreciate the reviewer comments that improve the quality of the manuscript.
Round 2
Reviewer 1 Report
Thanks to the authors for their comments and edits.
The authors reworked the manuscript.
However, the manuscript must be prepared with greater interest.
Please check the manuscript according to the MDPI template.
I ask authors to focus on:
1) They created a list of abbreviations.
2) Check the formatting of tables - margins (Table 2, 3, 4, 5, 6, 7).
Some tables are on two pages.
3) Check the decimal point markings.
4) Figure 11 - The font (text) in the picture is small. Enlarge the text.
5) The reference part contains the text from the template MDPI - delete lines 520-542
After editing, the manuscript will have sufficient information value and visual.
Author Response
Thanks to the authors for their comments and edits.
The authors reworked the manuscript.
However, the manuscript must be prepared with greater interest.
Please check the manuscript according to the MDPI template.
I ask authors to focus on:
- They created a list of abbreviations.
A list of abbreviations has been created, nevertheless since the template of MDPI does not include a list of abbreviations and it has been included as an Appendix.
2) Check the formatting of tables - margins (Table 2, 3, 4, 5, 6, 7).
Some tables are on two pages.
The tables have been placed in one page. Regarding the width of the Tables, the authors suggest that the editor change the orientation of the page (landscaped) so that any of the tables do not exceed the limits of the page, if is needed.
3) Check the decimal point markings.
Done
4) Figure 11 - The font (text) in the picture is small. Enlarge the text.
Done
5) The reference part contains the text from the template MDPI - delete lines 520-542
Done
After editing, the manuscript will have sufficient information value and visual.
The authors really appreciate the comments of the reviewer for improving the quality of the paper.
Reviewer 2 Report
The authors adequately addressed all issues raised by the reviewer. The authors also corrected several linguistic errors. However, another aspect still deserves the attention of the Authors: the use of “,” instead of “.” for decimals. Sometimes the Authors only used the “,” (Figure 6, Figure 9, Figure 10, Figure 12) or “,” in combination with “.” (Table 4) for decimals. Please use the “.” symbol for all decimals.
Author Response
The authors adequately addressed all issues raised by the reviewer. The authors also corrected several linguistic errors. However, another aspect still deserves the attention of the Authors: the use of “,” instead of “.” for decimals. Sometimes the Authors only used the “,” (Figure 6, Figure 9, Figure 10, Figure 12) or “,” in combination with “.” (Table 4) for decimals. Please use the “.” symbol for all decimals.
The "," for the decimals have been corrected.
The authors really appreciate the comments and of work of the reviewer for improving the quality of the paper.